# Single-Step IGHV Next-Generation Sequencing Detects Clonality and Somatic Hypermutation in Lymphoid Malignancies: A Phase III Diagnostic Accuracy Study

**DOI:** 10.3390/cancers15184624

**Published:** 2023-09-19

**Authors:** Anna Gazzola, Mohsen Navari, Claudia Mannu, Riccardo Donelli, Maryam Etebari, Pier Paolo Piccaluga

**Affiliations:** 1Hematopathology Unit, IRCCS Azienda Opedaliera-Universitaria di Bologna S. Orsola-Malpighi, 40138 Bologna, Italy; anna.gazzola@unibo.it (A.G.); claudia.mannu@aosp.bo.it (C.M.); 2Department of Medical Biotechnology, School of Paramedical Sciences, Torbat Heydariyeh University of Medical Sciences, Torbat Heydariyeh 95196-33787, Iran; mohsen.navari@gmail.com; 3Research Center of Advanced Technologies in Medicine, Torbat Heydariyeh University of Medical Sciences, Torbat Heydariyeh 95196-33787, Iran; 4Bioinformatics Research Center, Mashhad University of Medical Sciences, Mashhad 91779-48564, Iran; 5Biobank of Research, IRCCS Azienda Opedaliera-Universitaria di Bologna, 40138 Bologna, Italy; riccardo.donelli@studio.unibo.it; 6Department of Medical and Surgical Sciences, Institute of Hematology and Medical Oncology “L&A Seràgnoli”, Bologna University School of Medicine, 40126 Bologna, Italy; 7Health Sciences Research Center, Torbat Heydariyeh University of Medical Sciences, Torbat Heydariyeh 33787-95196, Iran; maryam.etebary@gmail.com

**Keywords:** clonality, immunoglobulin heavy chain, BIOMED2, lymphoma, leukemia, evidence-based medicine, diagnostic accuracy, PCR, next-generation sequencing, LymphoTrack^®^ IGH assay, LymphoTrack^®^ IGH somatic hypermutation assay

## Abstract

**Simple Summary:**

Clonality testing and somatic hypermutation analysis performed on B-cell receptor encoding genes are the most widely used molecular assays for lymphoma diagnostics. Currently, PCR-based methods standardized by the BIOMED2 consortium are regarded as the gold standard. In the last few years, new approaches based on next-generation sequencing (NGS) have been proposed and validated in phase I–II studies. Here, we present the first phase III diagnostic accuracy study, evaluating an NGS-based protocol (LymphoTrack^®^ IGH assay, and LymphoTrack^®^ IGH somatic hypermutation assay) compared to the gold standard. We formally documented a high diagnostic accuracy providing a clinical validation of the assays.

**Abstract:**

Background: Multiplex PCR based on consensus primers followed by capillary electrophoresis and Sanger sequencing are considered as the gold standard method for the evaluation of clonality and somatic hypermutation in lymphoid malignancies. As an alternative, the next-generation sequencing (NGS) of immune receptor genes has recently been proposed as a solution, due to being highly effective and sensitive. Here, we designed a phase III diagnostic accuracy study intended to compare the current gold standard methods versus the first commercially available NGS approaches for testing immunoglobulin heavy chain gene rearrangements. Methods: We assessed IGH rearrangements in 68 samples by means of both the NGS approach (LymphoTrack^®^ IGH assay, and LymphoTrack^®^ IGH somatic hypermutation assay, run on Illumina MiSeq) and capillary electrophoresis/Sanger sequencing to assess clonality and somatic hypermutations (SHM). Results: In comparison to the routine capillary-based analysis, the NGS clonality assay had an overall diagnostic accuracy of 96% (63/66 cases). Other studied criteria included sensitivity (95%), specificity (100%), positive predictive value (100%) and negative predictive value (75%). In discrepant cases, the NGS results were confirmed by a different set of primers that provided coverage of the IGH leader sequence. Furthermore, there was excellent agreement of the SHM determination with both the LymphoTrack^®^ FR1 and leader assays when compared to the Sanger sequencing analysis (84%), with NGS able to assess the SHM rate even in cases where the conventional approach failed. Conclusion: Overall, conventional Sanger sequencing and next-generation-sequencing-based clonality and somatic hypermutation analyses gave comparable results. For future use in a routine diagnostic workflow, NGS-based approaches should be evaluated prospectively and an analysis of cost-effectiveness should be performed.

## 1. Introduction

According to the WHO classification, the characterization of B-cell non-Hodgkin lymphomas (B-NHLs) is usually based on clinical characteristics, cyto/histomorphology and immunophenotypes. However, 10–15% of cases represent a challenging diagnosis which necessitates further analysis such as clonality in order to confirm the initial suspicion [1,2,3,4,5]. Therefore, differentiating clonal B-cell populations by means of analysis of the immunoglobulin heavy chain (IGH) locus becomes a useful approach for the diagnosis of B-NHLs [1,2,3,4]. In fact, the evaluation of these markers provides useful information. First, IGH clonality assessment represents a solid support in the diagnostic definition of lymphoid disorders through which the origin of such tumors can be elucidated [2,3,6,7]. Second, the analysis of somatic hypermutation (SHM) in IGH genes confirms the classifications of B-cell lineage and facilitates the stratification of patients into different prognostic groups [8]. In particular, the biological relevance of SHM suggests that the immunoglobulin variable domain (IGV) SHM has clinical significance in different types of B-NHLs, which is supported by studies reported on splenic marginal zone lymphoma (SMZL), chronic lymphocytic leukemia (CLL) and mantle cell lymphoma (MCL) [9,10,11,12,13,14,15].

At present, different approaches can be used to detect the clonality and mutational status of IGH in B-NHLs: the most widespread is based on polymerase chain reaction (PCR) assays followed by capillary electrophoresis (CE) and/or Sanger sequencing. In these tests, multiplex PCR is followed by sequencing based on a series of consensus primers which cover most of the possible unique V(D)J rearrangements. In this manner, clonal proliferations can be detected with very high sensitivity and specificity [1,2]. To improve the clonality determination, in 2003, the standardized protocols and primers for multiplex PCR were developed by the BIOMED-2 group, through which the clonality detection was improved in terms of both efficiency and reproducibility [2]. This technique, while fairly fast and accurate, is however subject to inherent problems. Limited sensitivity caused by normal polyclonal background, pseudo-clonality, false-positive/negative results owing to lack of genetic material and the high subjectivity in the interpretation of results are all limitations of this technique [16,17]. 

In recent years, the study of lymphoid malignancies has been revolutionized by next-generation sequencing (NGS) technologies, changing the landscape of molecular knowledge [6,18,19,20,21,22,23]. These methods are expected to make complete genomic analysis cost effective, and thus feasible in routine clinical diagnostics. In particular, the deep sequencing of immune receptor gene populations would allow for molecular characterization with increased specificity and sensitivity for the detection of desired sequences, allowing researchers to better classify, stratify and monitor lymphoid neoplasms [18,19,20,21,22,23].

Consequently, novel approaches in DNA sequencing are expected to shed more light upon the complex topic of genetic rearrangement, especially through considerable improvement in the sensitivity and specificity for the tracking of monoclonal B cell expansions, and aid to elucidating the clinical and molecular features which underlie the progression of certain cellular clones. 

In an effort to re-evaluate IGH rearrangement testing (clonality and mutation analysis) in the context of NGS technology, we have designed a phase III diagnostic accuracy study [24] with the aim of comparing the value of the first commercially available kits for NGS analysis (Invivoscribe Technologies’ LymphoTrack^®^ IGH assay and LymphoTrack^®^ IGH somatic hypermutation assay, both of which are formatted for Illumina MiSeq instruments) with the gold standard analysis based on capillary electrophoresis discrimination using BIOMED2 primers.

Our goal was to test a new technique that is able to reduce interpretive subjectivity, improve effectiveness in detecting clonal populations in B-NHLs and facilitate a simplified one-step evaluation of somatic hypermutation testing in CLL samples.

## 2. Materials and Methods

### 2.1. Study Design and Case Selection

We designed a study to assess the efficacy of new potential diagnostic tests based on NGS technology, namely LymphoTrack^®^ IGH assay and LymphoTrack^®^ IGH somatic hypermutation assay formatted for the Illumina MiSeq instruments (Invivoscribe Technologies, Inc., San Diego, CA, USA) compared with a reference standard BIOMED2 PCR-based method [2]. This study was designed to fulfill the standards for the reporting of diagnostic accuracy studies (STARD) statement (http://www.stard-statement.org, accessed on 4 July 2014).

We analyzed 68 samples of B-lymphoproliferative disorders, including 51 CLL, 8 follicular lymphoma (FL), 3 MCL and 6 reactive lymphoid hyperplasia (RLH), collectively obtained from peripheral blood (PBL) (N = 51) or formalin-fixed paraffin-embedded (FFPE) tissue (lymph nodes, N = 17) (Table 1). The diagnosis was performed based on morphology, immunophenotype, FISH and molecular and clinical information, according to the WHO classification of tumors of the hematopoietic and lymphoid tissues [5]. 

The positive control consisted of DNA extracted from a B-cell line with a known, well-characterized IGH rearrangement, while the negative control consisted of tonsil DNA characterized by no sequence with a frequency higher than 1%.

All subjects gave their informed consent to molecular diagnostics. The study was designed and conducted according to the evidence-based medicine rules, respecting the QUADAS, REMARK, and STARD requirements (Appendix A).

### 2.2. DNA Extraction

We extracted genomic DNA using the QIAamp DNA kit (Qiagen Limburg, The Netherlands) and analyzed it for purity and concentration using a NanoDrop spectrophotometer (Thermo Scientific, Wilmington, DE, USA) and Qubit^®^ 2.0 Fluorometer (Life Technologies Carlsbad, CA, USA) according to the manufacturers’ instructions. The extracted DNA samples were considered eligible based on the following criteria: a 260/280 nm ratio between 1.8 and 2.0, and a 260/230 ratio of about 2.2. As an internal control to verify the DNA integrity, the amplification of a multiplex PCR control (gene segments from 100 to 400 bp) (Specimen Control Size Ladder—Invivoscribe Technologies, Inc., San Diego, CA, USA) was performed in every sample, according to the manufacturer’s instructions. 

### 2.3. IGH Clonal Analysis by PCR/Capillary Electrophoresis

IGH-FR1 rearrangements were evaluated in 68 samples by means of traditional capillary electrophoresis methods using family-specific VH primers in combination with one JH consensus primer, according to the EuroClonality guidelines (http://www.euroclonality.org/, accessed on 4 July 2014) [2,17]. An automated thermocycler (Eppendorf, Hamburg, Germany) was used for DNA amplification. Each 50 microliter PCR reaction included 100 ng of template DNA and 1 U of Ampli-Taq Gold DNA polymerase (Life Technologies Carlsbad, CA, USA). The cycling parameters were as described in the kit protocol, and as follows: pre-activation for 7 min at 95 °C, 35 cycles of: denaturation (95 °C for 30 s), annealing (60 °C for 30 s), extension (72 °C for 60 s), followed by a final extension at 72 °C for 10 min. The PCR resulted in products ranging in size from 310 to 360 bp. Two separate reactions of amplification were applied to all samples, which were further resolved using capillary electrophoresis. For this purpose, 1 µL of PCR product with 0.5 µL of a standard molecular weight product (LIZ—Life Technologies, Carlsbad, CA, USA) was mixed with 12 µL of formamide to induce denaturation for 1 min at 95 °C. Subsequently, every sample was run on an ABI Prism 310 capillary electrophoresis instrument (ThermoFisher, Carlsbad, CA, USA) to determine the clonal character according to the EuroClonality guidelines.

### 2.4. IGH Hypermutation Analysis by Sanger Sequencing

IGH FR1 PCR products were directly sequenced in 51 clonal samples to establish the mutational status of IGH. Briefly, ExoSAP (GE Healthcare Life Sciences, Piscataway, NJ, USA) was applied to eliminate unincorporated primers and dNTPs. The sequencing reaction was prepared as follows: 1 µL of the treated PCR amplicons, 3.2 pM of sequencing primers and 1 µL of BigDye Terminator Mix (Life Technologies, Carlsbad, CA, USA) to a final volume of 20 µL. A total of 25 cycles of sequencing reactions were performed according to the standard protocol recommended by Life Technologies for 25 cycles. 

DNA purification was achieved using the DyeEX Spin Kit (Qiagen, Limburg, Netherlands), and DNA was re-suspended in 12 µL of Hi-Di formamide (Life Technologies, Carlsbad, CA, USA). The products were denatured at 95 °C for 3 min, chilled on ice and were fractionated on the ABI310 Genetic Analyzer (Life Technologies, Carlsbad, CA, USA) using the POP4 polymer (Life Technologies Carlsbad, CA, USA), and the electropherograms were evaluated with FinchTV 1.4 software (http://geospiza.com/finchtv, accessed on 4 July 2014, Geospiza Inc., Seattle, WA, USA). We used an online tool, namely IMGT (the international ImMunoGeneTics information system, http://www.imgt.org, accessed on 4 July 2014) to analyze IGH alignment and mutational status.

### 2.5. Library Preparation and Sequencing on MiSeq Platform 

DNAs from the 68 samples were tested with the LymphoTrack^®^ IGH assay, 51 of which were also tested with the LymphoTrack^®^ IGH somatic hypermutation assay (Invivoscribe Technologies, Inc., San Diego, CA, USA). The IGH FR1 master mixes were used in 68 samples to amplify from the Framework 1 (FR1) to the J region, which encompasses portions of the IGH FR1 region to the downstream J region. Furthermore, the leader master mixes were used in 28/68 cases to amplify the genomic DNA between the upstream leader (VHL) region and the downstream joining (J) region of the IGH gene, which spans the entire variable (V) region, including the FR1, CDR1, FR2, CDR2, FR3 and CDR3 regions. Specifically, VH leader primers were used for the IGH Somatic Hypermutation Assay. All the primers were designed with the aim of sequencing of PCR products on the MiSeq instrument. There were 24 leader master mixes and 24 FR1 master mixes with individual sequence indices to facilitate the examination of up to 24 different specimens per run. Genomic DNA was amplified with either leader master mixes or FR1 master mixes or both. To purify amplicons, we used Agencourt^®^ AMPure XP (Beckman Coulter, Indianapolis, IN, USA) before pooling the sequence libraries. The quantification of amplicons and libraries was achieved by means of a KAPA library quantification kit (KAPA Biosystems, Wilmington, MA, USA). Two kits, i.e., MiSeq Reagent kit v2 (500-cycle) and MiSeq Reagent kit v3 (600-cycle) (Illumina, San Diego, CA, USA) on a MiSeq (Illumina, San Diego, CA, USA) were used for the sequencing of the FR1 and leader master mixes, respectively. The resulting FASTq files were analyzed as described below.

### 2.6. IGH Clonality and Hypermutation Analysis by NGS

The FASTq data output from the MiSeq data was analyzed by LymphoTrack^®^ MiSeq software (version: 2.4.3) (included with the commercial assay kit) running on a Windows PC. For each sample, the software determined the DNA sequence, the V-J assignment, raw sequence counts, frequency of rearranged IGH, the degree of SHM determining if the sequence was in-frame, the presence of stop codons and the % coverage of the canonical V gene sequence (Figure 1A,B).

To analyze the clonality, the set of unique reads was aligned by BLAST (v2.2.28) against a reference database composed of IGH-V and IGH-J genes. The top scoring alignment for each V-gene and J-gene was assigned to the read, and the clonality percentage was calculated as the count of all the collapsed reads of that unique sequence divided by the total number of reads found to have a V and J primer sequence. 

We established criteria for assigning clonality based on parameters derived from both the relative and absolute frequencies of the reads. For samples with a minimum of 10,000 reads, a cutoff of 5% was used as an indication of clonality, while a cutoff of 2.5% was used to indicate clonality for samples with a minimum of 20,000 reads. Samples with less than 10,000 reads were deemed non-evaluable (Table 1). 

SHM status was determined in two steps. Alignment statistics obtained from the BLAST algorithm for the top clonal read allowed the calculation of mismatches and gaps to the V-gene reference sequence. The mutation frequency was calculated as the sum of mismatches and gaps divided by the effective V-gene length. To validate this calculation, the top clonal read was used as input to the IMGT/V-Quest web tool (International ImMunoGeneTics Information System, http://www.imgt.org, accessed on 4 July 2014) to calculate the percentage of identity, and the inverse of this percentage was taken as the mutation frequency.

### 2.7. Statistical Analysis

We used the CATmaker software (Centre for Evidence-Based Medicine, Oxford University, http://www.cebm.net, accessed on 4 July 2014) for the calculations of sensitivity (ST), specificity (SP), positive predictive value (PPV) and negative predictive value (NPV). In all of the analyses, the cut-off for significance was *p* ≤ 0.05. The correlation between the different tests was calculated using the Pearson’s correlation method and linear regression analysis. Relations were regarded as significant for R^2^ > 0.50 and for the Pearson correlation with *p* ≤ 0.05.

## 3. Results

### 3.1. Clonality Assessment by Classical and NGS Analysis

The clonality interpretation was carried out in agreement with the EuroClonality guidelines (http://www.euroclonality.org/, accessed on 4 July 2014): samples were considered as clonal if one or two reproducible peaks were observed and oligoclonal when multiple reproducible peaks (≥3) were present, while a Gaussian distribution of peaks was interpreted as a polyclonal population [2,17]. 

Particularly, in this cohort of 68 cases examined using conventional analysis, 57 clonal, 3 oligoclonal and 8 polyclonal samples were found (Table 2). 

Using LymphoTrack^®^ NGS assays, all MiSeq runs met the following run validity criteria: Q30 > 75% for 500 cycle, >70% for 600 cycle, cluster density greater than 600 K/cm^2^, PF > 90% and the total number of reads per run was greater than 10 million. 

Of the 68 samples tested, 2 samples were not evaluable by NGS due to a low number of reads (<10,000). A total of 54 samples were identified as clonal, 11 were identified as polyclonal and only 1 sample was identified as oligoclonal (Table 1). 

Representative clonal and polyclonal cases are shown in Figure 2A–L. 

For the purposes of determining the diagnostic efficiency of the molecular tests, we compared NGS with standard analysis: 8 out of 8 polyclonal samples and 54 out of 57 clonal samples had a concordant result. Three samples called clonal by CE analysis were identified as polyclonal by NGS. For example, the NGS and CE results of one of those cases are presented in Figure 2F,L. The oligoclonal sample was in agreement with the CE results. The overall diagnostic accuracy of NGS in comparison to CE was 95% ST, 100% SP, 100% PPV and 75% NPV (Table 3). 

### 3.2. Somatic Hypermutation Detection by Classical and NGS Analysis

Somatic hypermutation assessment was performed on 51 CLL samples for which clonality had been established using both conventional and NGS assays. Using the capillary electrophoresis/Sanger sequencing approach, 23 samples were determined to have a germline IGH configuration, 21 samples with a mutation frequency >2.0% and 7 samples not evaluable probably due to the consistent polyclonal background (Table 4).

Using the NGS testing, for the classification of a clonal sequence as ‘highly mutated’ the SHM rate must be ≥2%, it must be in frame, have no stop codons and show a high degree of coverage to the canonical V region (>95%). Using this approach and by means of IGH FR1/JH master mixes, 25 samples were determined to have germline IGH sequences and 26 samples had a mutation frequency > 2.0% (Table 4). These NGS results were confirmed by an independent mutational analysis performed by NGS with leader/JH master mixes on 28 samples for which enough DNA was available. The results showed 16 samples with a germline IGH configuration and 12 samples with a mutational spectrum > 2.0% (Table 4).

The alignment analysis results obtained via NGS IGH FR1/JH master mixes (N = 51 cases) and the results obtained with Sanger sequencing, performed to classify the V (variable), D (diversity) and J (joining) fragment recombination of IGH, showed a correlation of 86% (38 out of 44 agreement on VDJ assignments) (Table 4). The alignment analysis performed comparing Sanger sequencing and NGS leader/JH master mixes (N = 28 cases) data showed a correlation of 82%. Specifically, in five cases (Sample 21, 29, 40, 41 and 48) NGS and Sanger sequencing identified different gene families. However, NGS was consistent in the two assays (leader vs. FR1). In only one case (Sample 32) the IGH leader result agreed with the Sanger result, but the IGH FR1 identified a different VDJ family. 

All the cases (Sample 8, 12, 16, 24, 25, 46 and 50) that could not be evaluated by Sanger sequencing were detected by NGS IGH FR1 with four cases (Sample 16, 25, 46 and 50) identified as mutated and three cases (Sample 8, 12 and 24) identified as unmutated.

In total, there was excellent agreement of the SHM determination with both the LymphoTrack^®^ FR1 and leader assays compared to the IMGT analysis (Figure 3A,B). In addition, the two assays used by NGS (IGH FR1 assay and IGH leader assay) demonstrated excellent concordance. The identification of top evaluable clonal reads from MiSeq sequencing with both the leader and FR1/JH master mixes displayed a 96% sequence identity (27 out of 28) between the top evaluable clonal reads (Figure 4). In one case (Sample 32), different families were identified, but Sanger sequencing confirmed what was observed by leader analysis at NGS. 

On the other hand, when mutation frequency obtained by Sanger sequencing and NGS assays were compared, we documented an 84% (37 out of 44) concordance between Sanger sequencing and the IGH FR1 assay (Table 4 and Figure 5).

## 4. Discussion

Interrogation of the antigen receptor loci is an established, integral part of the routine diagnostic work up of a substantive percentage of cases for hematology and hematopathology laboratories. Identification of IGH clonality in NHL and assessment of somatic hypermutation frequency of IGHV regions in chronic lymphocytic leukemias are often requested for disease diagnosis and characterization. The first methods used to identify clonal rearrangements included restriction fragment, Southern blot hybridization (RF-SBH) techniques. These methods, however, suffered from disadvantages such as being labor-intensive and cumbersome, requiring large quantities of DNA besides not being appropriate for the analysis of many of the antigen receptor loci with less diversity. Over time, PCR-based clonality tests have replaced RF-SBH assays and are considered as the current gold standard method in most centers. The identification of clonality using PCR-based assays relies upon the over-representation of amplified V-D-J (or incomplete D-J) products, recognizable by gel electrophoresis [2,6,15].

As standardization is a basic requirement for diagnostic testing, in recent years, international consortia such as BIOMED2 have provided standardized PCR-based protocols for IGH, IGK and TCR analysis [2]. These approaches have proved quite useful, characterized by their efficiency, speed and cost-effectiveness for routine practice, represented by, for example, their sensitivity and need of minor DNA amounts. However, they retain some drawbacks. First, these assays are limited in discriminating clonal populations and multiple rearrangements which might be manifested as a single-sized peak. Furthermore, products that generate non-uniform Gaussian distributions are difficult to interpret, especially where amplicon products obtained from the IGK and TCR loci are concerned. In addition, the design of these assays fails to recognize the specific V-D-J DNA sequences which are needed to be tracked and analyzed subsequently. This is particularly important, since after the identification of clone-specific DNA sequences, the related clonal cell populations can be tracked based on these specific sequences in order to detect minimal residual disease. Additionally, minor clones possibly present at diagnosis can be missed using less sensitive capillary-based methods, making the identification of recurrent minor clones in the clonal evolution of the disease impossible.

Recently, the usage of NGS technologies has found more applications in clinical pathology laboratory, for example in clonality assessment [25,26,26,27,28]. Importantly, these technologies represent a fundamental advantage over currently used PCR-based methods: the possibility to recognize clonal populations with considerably low levels and the lower copy number detection limit which is theoretically determined only by Poisson sampling. 

In this study, we explored the ability of novel NGS-based assays for the detection of IGH clonal rearrangement and somatic hypermutation by comparing the first commercial NGS assays to the current gold standard capillary-based assay and Sanger sequencing. Although other studies have been published in this context [25,26,26,27,28], to our best knowledge, this is the first study performed fulfilling the requirements of a phase III diagnostic accuracy study. According to the STARD and QUADAS guidelines, all cases were analyzed with both methods, and the results of each test were blinded to the researchers performing the other test. There are other aspects which make our research unique. For example, Lay et al. used Ion Torrent for the detection of clonality in CLL using the LymphoTrack^®^ IGH assay but used an Ion Torrent S5 instrument [29]. Other studies used LymphoTrack^®^ on a MiSeq instrument, but instead focused on acute lymphoblastic leukemia (B-ALL) [30] and multiple myeloma (MM) [31,32]. Notably, in our hands, the LymphoTrack^®^ IGH assay demonstrated a remarkable sensitivity (95.0%) and specificity (100%) and therefore seems to have a high potential for clinical usage. Similar agreement between the classic and NGS methods for clonality detection in CLL [29], B-ALL [30] and MM [31,32] has been reported. Importantly, this assay could detect clonal patterns in samples with partial DNA degradation (e.g., FFPE tissues) and in cases for which relatively low DNA amounts were available. In our study, however, it is noteworthy that all discrepancies were recorded when the starting material was a FFPE lymph node, while fresh blood always gave identical results. Similarly, the non-evaluable cases due to the scan material were FFPE. Since, at present, most of lymphoma diagnoses are made on FFPE, this should be kept in mind when choosing the molecular approach. Three samples identified as clonal by CE analysis were found to be polyclonal based on NGS, one of which was from a reactive non-malignant lymphoid hyperplasia. It is possible that the other two discordant samples were the result of the lower resolution of the CE method, since in this assay all sequences (amplicons) of the same size will be detected as a single peak. Often within that single peak there will be several individual distinct DNA sequences. NGS analysis allows the distinction of each individual DNA sequence for a much greater resolution of the individual IGH gene rearrangements. On the other hand, since different IGHV and IGHJ primers are used, this may account for the discrepancy; particularly, it may be that a primer does not bind to a specific IGHV gene, or that in mutated IGHV rearrangements a primer can no longer bind efficiently because of mutations at the primer binding sites.

Despite the rigorous standardization of PCR tests, for example as suggested by the EuroClonality Group, the usage of these tests still faces a major complication, due to the significant degree of interpretive subjectivity [17,33,34]. The NGS-based analysis of samples might overcome these issues, offering the possibility of quantitative sequencing and thus proving a more objective characterization of the single clones included in a given sample. Furthermore, the ability of NGS-based approaches to identify the exact clone-associated sequence in each single experiment can also facilitate the subsequent analysis of minimal residual disease.

In addition, we provide evidence proving that an NGS-based assay is at least as effective as Sanger sequencing for detecting SHM. In particular, the LymphoTrack^®^ IGH assay and LymphoTrack^®^ IGH somatic hypermutation assay were shown to be both robust and reproducible, with comparable results when leader or FR1 primers were used. In addition, we documented a remarkable correspondence between LymphoTrack^®^ and Sanger sequencing results. Notably, cases with a significant polyclonal background were easily resolved by LymphoTrack^®^, while they required single band isolation and re-sequencing and sometimes even resulted in interpretative failure, when analyzed by conventional Sanger sequencing. From this perspective, the NGS-based approach appeared to be far superior to the current gold standard method. Similarly, Lay et al. reported NGS to be more robust for the detection of IGH somatic hypermutation than PCR fragment analysis in CLL patients.

In a few instances, however, we obtained discrepant results concerning the identification of the specific IGH families, which may affect the clinical decision. First, we observed one case in which IGH FR1 and leader assays led to different results, while the Sanger sequencing was comparable to the leader assay. In this regard, it should be noted that the VHL/J (leader) master mix covers the whole VH1 region and can detect clonality and mutations which might be missed by FR1/J master mixes (Table 2, Sample 60, missed by FR1 but picked by leader). However, the MiSeq reagents for the leader are currently more expensive (about 30% more) and the reaction takes longer (by approximately 16 h). A comparison of IGH VHL/J and IGH FR1/J master mix performance is listed in Table 5. 

In five cases, furthermore, Sanger sequencing data were different from data obtained using NGS. However, the NGS somatic hypermutation result was confirmed with a second independent assay that targeted and amplified the leader region. In these cases, the study design (phase III diagnostic accuracy) necessitated calling in favor of the current gold standard method. However, as the two different chemistries used for NGS led to the same result and NGS theoretically provided a higher specificity (the length of the analyzed DNA fragments being longer), it is more likely that NGS offered the correct result. Unfortunately, we could not re-analyze the cases by Sanger sequencing using a different set of primers (i.e., leader) due to the lack of residual DNA.

Notably, NGS, in general, might offer additional opportunities in diagnostics. Clonality can also be studied through RNA sequencing and more extensive DNA sequencing. This may also allow for the detection of cancer-associated mutations and further enhance the diagnostic utility of the assay. Such data were not available for our series; indeed, it will be worthwhile to compare clonality and whole exome sequencing, as well as RNA sequencing, to better define their possible role in diagnostics. In this regard, NGS also offers a direct interpretation of double rearrangements to distinguish bi-clonal cases from ones with a nonproductive rearrangement. We did not observe any in the present series, but this is certainly one of the advantages offered by the new technology.

NGS has also been proposed because it is capable of providing a solid tool for minimal residual disease monitoring. Since all rearrangements are identified, they can be monitored over time in serial samples. This also allows us to recognize and anticipate a possible divergent, clonally unrelated relapse, in contrast to conventional methods. It should be noted, however, that the sensitivity of NGS is strictly related to the target amount and therefore the DNA input). It must be further tested, despite good initial evidence, whether this can overcome the performance of quantitative and especially digital PCR [35]. 

Finally, a relevant issue for proposing NGS-based approaches in the clinical routine is the final cost of the assay. Specifically, the direct costs for reagents are usually, at least at present, higher for the NGS tests. On the other hand, however, when the classical tests are used, the analysis of IGH loci includes sequence recognition and mutational assessment that would require additional expenditure. Therefore, depending on the specific aim of the analysis, NGS might already be regarded as cost effective. In addition, by offering a more objective interpretation and, in the case of LymphoTrack^®^, a simplified protocol for sample preparation and included bioinformatics software and analysis, costs related to operators can be minimized. In particular, it would be ideal to use NGS-based systems when molecular testing is performed in centralized laboratories, as the costs for sequencing are definitely reduced by multiplexing. On the other hand, if fewer samples have to be analyzed and information on sequence recognition or SHM status is not required, BIOMED2 and similar approaches may still be more convenient and cost effective.

## 5. Conclusions

In conclusion, LymphoTrack^®^ was determined to be as effective, if not more so, than conventional PCR-based methods in the identification of IGH clonality and SHM, with advantages in special cases such as those with a significant polyclonal background. Perspective phase IV studies evaluating the cost-effectiveness of the test and its impact on clinical decisions are now warranted in order to further define its role in routine diagnostics.

## Figures and Tables

**Figure 1 cancers-15-04624-f001:**
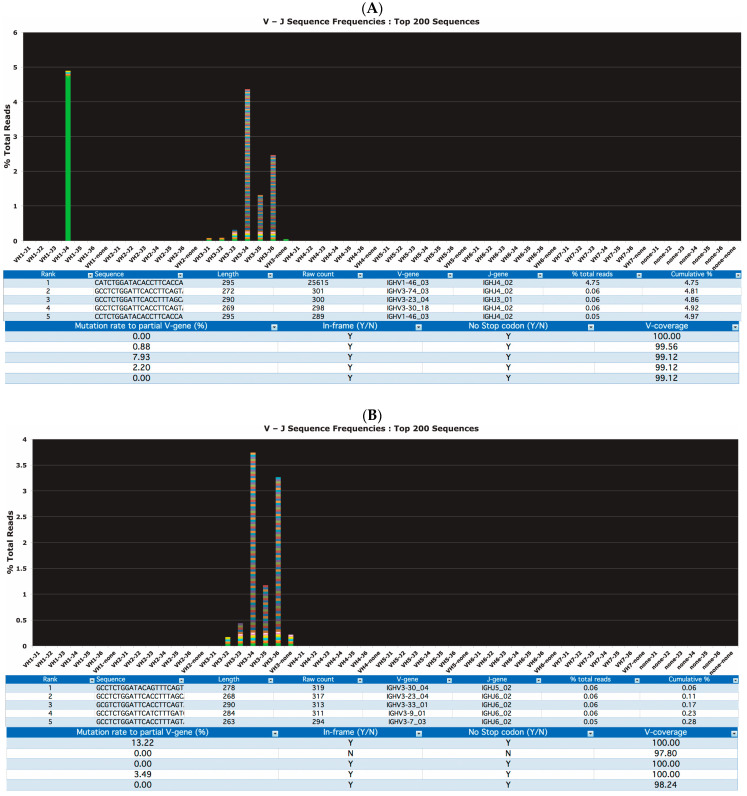
Representative clonal (**A**) and polyclonal (**B**) cases visualized by LymphoTrack^®^ MiSeq software. The software determined the DNA sequence, the V-J assignment, raw sequence counts, frequency of rearranged *IGH*, degree of somatic hypermutation (SHM), presence of stop codons and the % coverage of the canonical V gene sequence.

**Figure 2 cancers-15-04624-f002:**
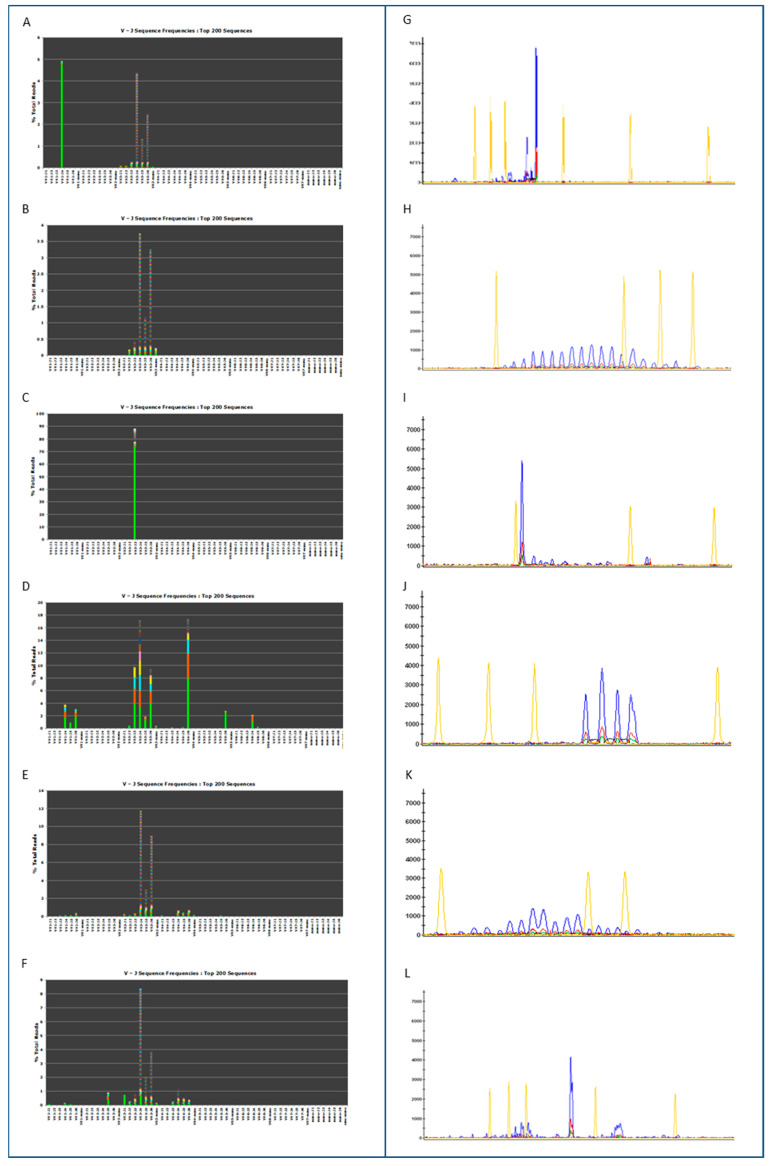
Representative clonal and polyclonal cases by NGS (**A**–**F**) and **BIOMED**-2 PCR (**G**–**L**). (**A**): Positive control (538,906 total reads). (**B**): Negative control (558,864 total reads). (**C**): Positive sample (#4) (545,858 total reads). (**D**): Oligoclonal sample (#57) (18,056 total reads). (**E**): Polyclonal sample (#61) (744,191 total reads). (**F**): Discordant sample (#54) (822,638 total reads). (**G**): Positive control; (**H**): Negative control (**I**): Positive sample (#4). (**J**): Oligoclonal sample (#57). (**K**): Polyclonal sample (#61); (**L**): Discordant sample (#54).

**Figure 3 cancers-15-04624-f003:**
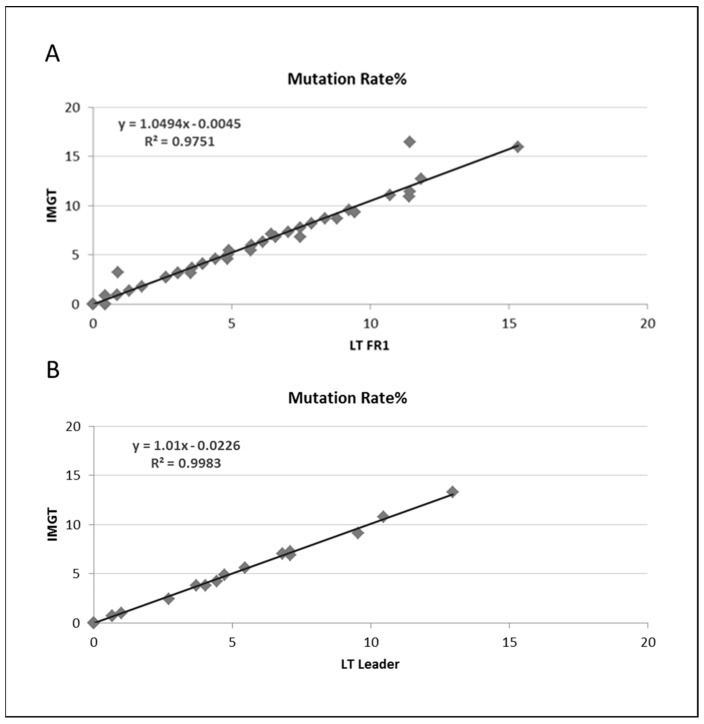
(**A**) Comparison of somatic hypermutation (SHM) rates determined by LymphoTrack^®^ FR1 with LymphoTrack^®^ software versus LymphoTrack^®^ FR1 with IMGT for 68 samples. (**B**) Comparison of SHM rate determined by LymphoTrack^®^ FR1 with LymphoTrack^®^ software versus LymphoTrack^®^ Leader with LymphoTrack^®^ software for 28 samples (LymphoTrack^®^ = LT in the figure).

**Figure 4 cancers-15-04624-f004:**
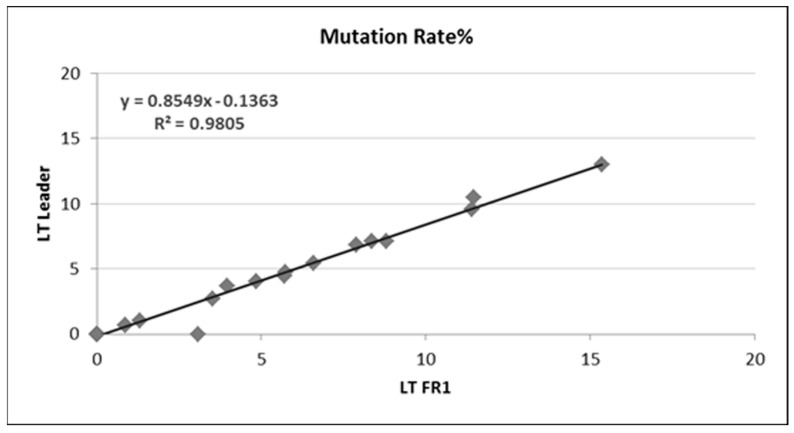
Comparison of somatic hypermutation (SHM) rates determined by LT FR1 with LT software versus LT leader with LT software for 28 samples.

**Figure 5 cancers-15-04624-f005:**
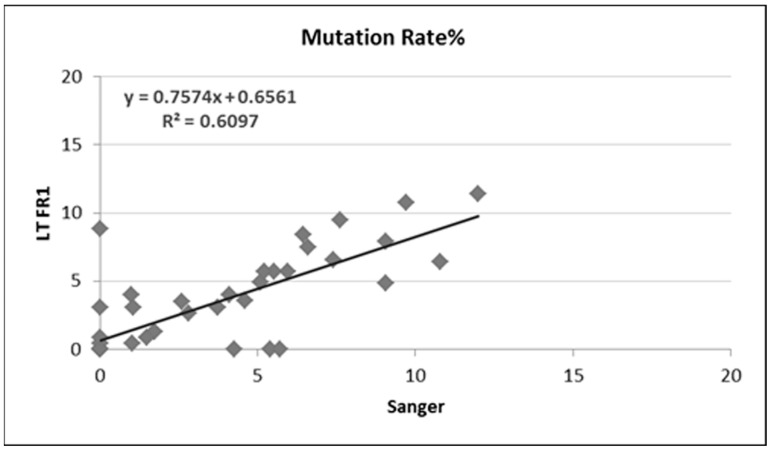
Comparison of somatic hypermutation (SHM) rates determined by PCR/Sanger versus LT FR1 for 44 CLL samples.

**Table 1 cancers-15-04624-t001:** Power of detecting clonality by NGS.

Frequency of Clonal Population	Margin of Error	Power *	
		95%	99%
1.0%	0.1% (0.9–1.1%)	134,379	191,545
1.0%	1.0% (0–2%)	1809	2712
2.5%	0.5% (2–3%)	11,009	19,760
2.5%	1% (1.5–3.5%)	2933	5381
5.0%	0.5% (4.5–5.5%)	25,739	36,675
5.0%	1.0% (4–6%)	6688	9597

***** Number of reads required to identify the given percentage of clonal population with the given margin of error.

**Table 2 cancers-15-04624-t002:** Case description and comparison of the results of clonality analysis by NGS and conventional capillary electrophoresis assays.

N	Sample	Diagnostic	NGS	Capillary Electrophoresis
1	PBL	CLL	Clonal	Clonal
2	PBL	CLL	Clonal	Clonal
3	PBL	CLL	Clonal	Clonal
4	PBL	CLL	Clonal	Clonal
5	PBL	CLL	Clonal	Clonal
6	PBL	CLL	Clonal	Clonal
7	PBL	CLL	Clonal	Clonal
8	PBL	CLL	Clonal	Clonal
9	PBL	CLL	Clonal	Clonal
10	PBL	CLL	Clonal	Clonal
11	PBL	CLL	Clonal	Clonal
12	PBL	CLL	Clonal	Clonal
13	PBL	CLL	Clonal	Clonal
14	PBL	CLL	Clonal	Clonal
15	PBL	CLL	Clonal	Clonal
16	PBL	CLL	Clonal	Clonal
17	PBL	CLL	Clonal	Clonal
18	PBL	CLL	Clonal	Clonal
19	PBL	CLL	Clonal	Clonal
20	PBL	CLL	Clonal	Clonal
21	PBL	CLL	Clonal	Clonal
22	PBL	CLL	Clonal	Clonal
23	PBL	CLL	Clonal	Clonal
24	PBL	CLL	Clonal	Clonal
25	PBL	CLL	Clonal	Clonal
26	PBL	CLL	Clonal	Clonal
27	PBL	CLL	Clonal	Clonal
28	PBL	CLL	Clonal	Clonal
29	PBL	CLL	Clonal	Clonal
30	PBL	CLL	Clonal	Clonal
31	PBL	CLL	Clonal	Clonal
32	PBL	CLL	Clonal	Clonal
33	PBL	CLL	Clonal	Clonal
34	PBL	CLL	Clonal	Clonal
35	PBL	CLL	Clonal	Clonal
36	PBL	CLL	Clonal	Clonal
37	PBL	CLL	Clonal	Clonal
38	PBL	CLL	Clonal	Clonal
39	PBL	CLL	Clonal	Clonal
40	PBL	CLL	Clonal	Clonal
41	PBL	CLL	Clonal	Clonal
42	PBL	CLL	Clonal	Clonal
43	PBL	CLL	Clonal	Clonal
44	PBL	CLL	Clonal	Clonal
45	PBL	CLL	Clonal	Clonal
46	PBL	CLL	Clonal	Clonal
47	PBL	CLL	Clonal	Clonal
48	PBL	CLL	Clonal	Clonal
49	PBL	CLL	Clonal	Clonal
50	PBL	CLL	Clonal	Clonal
51	PBL	CLL	Clonal	Clonal
52	FFPE	FL	Clonal	Clonal
53	FFPE	MCL	Clonal	Clonal
54	FFPE	MCL	Polyclonal	Clonal
55	FFPE	FL	Polyclonal	Clonal
56	FFPE	FL	Clonal	Clonal
57	FFPE	FL	Oligoclonal	Oligoclonal
58	FFPE	MCL	* Not evaluable	Oligoclonal
59	FFPE	FL	* Not evaluable	Oligoclonal
60	FFPE	RLH	Polyclonal	Clonal
61	FFPE	FL	Polyclonal	Polyclonal
62	FFPE	FL	Polyclonal	Polyclonal
63	FFPE	FL	Polyclonal	Polyclonal
64	FFPE	RLH	Polyclonal	Polyclonal
65	FFPE	RLH	Polyclonal	Polyclonal
66	FFPE	RLH	Polyclonal	Polyclonal
67	FFPE	RLH	Polyclonal	Polyclonal
68	FFPE	RLH	Polyclonal	Polyclonal

* Two samples were not evaluable due to low amounts of amplifiable DNA and total reads < 10,000. B-CLL: B-cell chronic lymphocytic leukemia, FL: follicular lymphoma, MCL: mantle cell lymphoma, RLH: reactive lymphoid hyperplasia, FFPE: formalin-fixed paraffin-embedded tissue, PBL: peripheral blood lymphocyte.

**Table 3 cancers-15-04624-t003:** Diagnostic accuracy of the IVS NGS assay (FR1) for clonality detection—CATmaker software (Centre for Evidence-Based Medicine, Oxford University).

			**Capillary Electrophoresis**		
			**Clonal**		**Non-Monoclonal**
NGS	Clonal		54		0
	Non-monoclonal		3		9
**Parameter**		**Value**		**95% CI**	
ST		95%		89–100	
SP		100%		100–100	
Pre-test probability		86%		78–95	
PPV		100%		100–100	
NPV		75%		51–100	
LR+		Not evaluable		Not evaluable	
LR−		0.05		0.02–0.16	

ST: sensitivity, SP: specificity, PPV: positive predictive value, NPV: negative predictive value, LR+: likelihood ratio positive, LR−: likelihood ratio negative.

**Table 4 cancers-15-04624-t004:** *IGH* mutational analysis (NGS vs. Sanger). Case order is equal to Table 2.

N	Sanger			NGS-FR1			NGS-Leader		
	V	J	%Mut Rate	V	J	%Mut Rate	V	J	%Mut Rate
1	V1-69_13	J6_02	0	V1-69_13	J6_02	0.00	n/a	n/a	n/a
2	V3-74_03	J4_02	0	V3-74_01	J4_02	0.00	n/a	n/a	n/a
3	V1-18_01	J6_02	1.03	V1-18_01	J6_02	0.44	n/a	n/a	n/a
4	V3-33_03	J3_02	5.53	V3-33_01	J3_02	5.73	n/a	n/a	n/a
5	V3-15_07	J6_02	9.7	V3-15_02	J6_02	10.73	n/a	n/a	n/a
6	IV3-21_01	J4_02	0	V3-21_02	J4_02	0.00	n/a	n/a	n/a
7	V1-18_01	J6_02	0	V1-18_01	J6_02	0.00	n/a	n/a	n/a
8	NV	NV	NV	V4-34_02	J4_02	0.00	V4-34_01	J4_02	0
9	V5-a_04	J4_02	5.1	V5-a_03	J4_02	4.91	n/a	n/a	n/a
10	V3	J4_02	4.1	V3-30_02	J4_02	3.96	n/a	n/a	n/a
11	V1-69_13	J6_02	5.4	V1-69_13	J6_02	0.00	n/a	n/a	n/a
12	NV	NV	NV	V4-39_01	J5_02	0.00	V4-39_01	J5_02	0
13	V1-69	J3_02	0	V1-69_06	J3_02	0.00	n/a	n/a	n/a
14	V3	J5_02	6.6	V3-11_05	J5_02	7.49	n/a	n/a	n/a
15	V1-18_01	J4_02	0	V1-18_01	J4_02	0.44	n/a	n/a	n/a
16	NV	NV	NV	V3-21_02	J4_02	9.25	n/a	n/a	n/a
17	V3-72_02	J6_02	10.8	V3-72_01	J6_02	6.44	n/a	n/a	n/a
18	V4-34_13	J4_02	2.81	V4-34_02	J4_02	2.63	n/a	n/a	n/a
19	V1-2_03	J4_02	4.6	V1-2_04	J4_02	3.57	n/a	n/a	n/a
20	V3-21_02	J6_02	1.05	V3-21_02	J6_02	3.08	n/a	n/a	n/a
21	V3-21_02	J6_02	0	V1-2_04	J6_02	0.00	V1-2_04	J6_02	0
22	V3-9_02	J4_02	3.75	V3-9_01	J4_02	3.06	n/a	n/a	n/a
23	V3-72_02	J3_02	7.6	V3-72_01	J3_02	9.44	n/a	n/a	n/a
24	NV	NV	NV	V3-49_05	J5_02	0.00	n/a	n/a	n/a
25	NV	NV	NV	V3-21_02	none	7.49	n/a	n/a	n/a
26	V5-a_04	J6_02	0	V5-a_03	J6_02	0.00	n/a	n/a	n/a
27	V4-34_02	J6_02	7.4	V4-34_02	J6_02	6.58	V4-34_01	J6_02	5.46
28	V3-21_02	J5_02	1.74	V3-21_02	J5_02	1.32	V3-21_01	J5_02	1.01
29	V3-21_02	J5_02	0	V3-7_01	J1_01	8.81	V3-7_01	J1_01	7.09
30	V4-34_02	J5_02	5.94	V4-34_02	J5_02	5.70	V4-34_01	J5_02	4.44
31	V3-30_04	J4_02	0	V3-30_04	J4_02	0.00	V3-30_04	J4_02	0.00
32	V4-34_01	J4_02	0	V3-48_02	J3_02	3.08	V4-34_01	J4_02	0.00
33	V4-4_02	J6_02	0	V4-4_02	J6_02	0.00	V4-4_02	J6_02	0.00
34	V4-59_01	J6_02	0	V4-59_01	J6_02	0.00	V4-59_01	J6_02	0.00
35	V3-74_03	J4_02	2.6	V3-74_03	J4_02	3.52	V3-74_03	J4_02	2.70
36	V1-69_01	J6_04	0	V1-69_13	J6_04	0.00	V1-69_01	J6_04	0.00
37	V3-21_01	J6_02	0	V3-21_02	J6_02	0.00	V3-21_01	J6_02	0.00
38	V1-69_13	J6_02	0	V1-69_13	J6_02	0.00	V1-69_01	J6_02	0.00
39	V4-34_03	J4_02	12	V4-34_02	J5_02	11.40	V4-34_01	J5_02	9.56
40	V3-9_02	J6-02	5.72	V1-69_13	J6_04	0.00	V1-69_01	J6_04	0.00
41	V3-21	J4_02	1	V3-48_03	J4_02	3.96	V3-48_03	J4_02	3.72
42	V4-34_02	J6_03	1.5	V4-34_02	J6_03	0.88	V4-34_01	J6_03	0.68
43	V3-74_03	J5_02	6.45	V3-74_03	J5_02	8.37	V3-74_03	J5_02	7.09
44	V1-69_13	J6_02	0	V1-69_13	J6_02	0.00	V1-69_01	J6_02	0.00
45	V3-33_01	J6_02	5.2	V3-33_01	J6_02	5.73	V3-33_01	J6_02	4.73
46	NV	NV	NV	V4-59_08	J5_02	15.35	V4-59_08	J5_02	12.97
47	V3-21_02	J4_02	0	V3-21_02	J4_02	0.88	V3-21_01	J4_02	0.68
48	V4-34_02	J6_02	9.05	V3-23_04	J4_02	4.85	V3-23_01	J4_02	4.05
49	V1-58_01	J6_03	4.27	V1-58_01	J6_03	0.00	V1-58_01	J6_03	0.00
50	NV	NV	NV	V3-23_04	J4_02	11.45	V3-23_01	J4_02	10.47
51	V4-34_02	J6_02	9.05	V4-34_02	J6_02	7.89	V4-34_01	J6_02	6.83

NV: not evaluable; n/a: not applicable.

**Table 5 cancers-15-04624-t005:** Comparison of VHL/J and FR1/J master mixes.

Characteristic	VHL/J	FR1/J
Amplicon size (bp)	500–570	290–360
PCR cycle	32	29
MiSeq Reagent kit	600–cycle	500–cycle
MiSeq run time (hr)	55	39
Pros	Whole VH region	Higher PCR efficiency Lower cost Faster results
Cons	Lower PCR efficiency Higher cost More time-consuming	Partial FR1 region

## Data Availability

Not applicable.

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
