# Peer review of "Single-Step IGHV Next-Generation Sequencing Detects Clonality and Somatic Hypermutation in Lymphoid Malignancies: A Phase III Diagnostic Accuracy Study"

_cancers, 2023, doi:10.3390/cancers15184624_

Round 1

Reviewer 1 Report

This is an interesting manuscript comparing innovative NGS based assays to traditional CE/Sanger sequencing for the diagnosis of B-Cell malignancies.  

Considering their increasing prevalence in the routine of pathology labs in multiple diseases, there is great value in understanding the sensitivity and specificity of NGS based approaches. Importantly, the reported results could push the translation of NGS appoaches not only in B-Cell lymphomas, but also in other hematological malignancies, such as T-Cell lymphomas.

The work presented here is well implemented and quite extensive, however it presents minor limitations that should be addressed before the publication in Cancers journal:

General points:

1. Did the authors have the chance to compare CE/Sanger and NGS based results with pre-clinical whole exome sequencing data (also in only few cases)? Since great concordancy is expected, WES data could be used to better assess clonality from different sources of material taking into account the cancer cell fraction, ultimately allowing to assess clonality. Also, WES data can be used for identification of canonical and new mutations (over normal), allowing a more precise diagnosis and stratification of lymphoid patients.

2. Did the authors test the different approaches (CE/Sanger vs NGSs) on diagnostic and relapsed samples? Did they detected the same clone? Any additional mutation on the founding clone/s? Any difference between CE/Sanger-NGSs appoaches?

3. The authors detected 3 patients as oligoclonal (#54-55-60) using NGS, but clonal by CE/Sanger. Is this explained by increased sensitivity of NGS assays? Could we detect the frequency of the different clones? Are some clones very rare? In case, could the authors comment on the potential of NGS assays increased capacity to identify relapsed/refractory patients in PLB, as well as the minimal residual disease (MRR) before then CE/Sanger the ?

4. Regarding the source of the samples: any DLBCL or B-ALL samples? Also, did the authors test fresh lymphonode material? Are FFPE samples coming from lymphonode? 

Author Response

Dear Sir

we first of all wish to thank you for the useful and constructive comments. We modified our manuscript according to all your advice

Please find details attached

Reviewer 2 Report

Gazzola and colleagues compared two methods to identify clonal B cell expansions in human diagnostic tissue or blood samples. They compared a well-established approach combining PCR amplification of rearranged IGHV genes from genomic DNA, followed by fragment length analysis of the amplificates (clonal rearrangements have an identical length, in polyclonal B cell samples, many bands of different length are obtained). In the second approach, rearranged IGHV genes are also amplified with sets of IGHV and IGHJ primers from genomic DNA with kits from Invivoscribe, but the amplificates are then undergoing deep sequencing and clonality is determined based on the IGHV sequences. The study included over 60 samples from B cell leukemias, lymphomas and reactive tissues. Overall, a very good concordance of the results was obtained. Whereas the classical method is principally faster, cheaper, and needs less bioinformatic expertise, the deep sequencing approach has the advantage that the IGHV gene used is identified and somatic mutations are detectable, so that more information about the lymphoma clone is obtained. Moreover, having identified the clonal IGHV gene sequence can aid in the design of minimal residual disease approaches to follow the clone during and after treatment.

Even though several similar studies have been performed, this direct comparison of the two approaches as part of a phase 3 study is valuable and informative.

Minor points - issues to be better discussed

1) In Figs. 1B and 2B, showing polyclonal samples, only products for rearrangements using members of the VH3 subgroup are visible (this is also true for the polyclonal background from normal B cells in Fig. 1A). VH3 is indeed the largest subgroup, used by about 40% of B cells, but it is worrisome that no rearrangements using genes of the other 6 groups (at least the also relatively frequently used VH1 and VH4) are visible. Was it controlled that all primers work? how can this be explained. This needs to be discussed.

2) There are a few examples where different clonal IGHV rearrangements were obtained with the two methods, or where with one approach only polyclonal rearrangements were seen, but with the other a clonal rearrangement was detected (Table 2). These discrepancies need to be better discussed. As the approaches use IGHV and IGHJ primers, it may to that a primer does not bind to a specific IGHV gene, or that in mutated IGHV rearrangements a primer can no longer bind efficiently because of mutations at the primer binding sites. Can this explain some of the discrepancies? Moreover, about 10-20% of human B cells and B-cell malignancies carry two IGHV rearrangements (typically one productive and one non-productive one). Where never two clonal rearrangements obtained? In CLL, the largest group of samples studied here, there is even data that a fraction of CLL carries two productive IGHV rearrangements (e.g., Am J Hematol 2013;88(4):277-82). Was this never observed here?

3) The comparison of the two methods provides identical results for the 51 samples of CLL. This is not surprising, as the DNA quality from peripheral blood samples is very high, often a very high frequency of malignant CLL cells is seen, and many cases have unmutated IGHV genes, so that mutations at primer binding sites are less a problem for false negative results (clonal rearrangement missed).  However, among the 11 cases with FFPE material (impaired DNA quality) and low-grade lymphomas (FL, MCL) (Table 2), there are 2 discrepant cases and two where the NGS approach was not evaluable (plus one reactive lymph mode with a discrepant result). So the "excellent agreement" for the two methods is only seen for the CLL blood samples, but less so for the FFPE lymphoma samples. This need to be more critically discussed.

Minor points - formating and language issues

a) In the mutation field, mutation rate typically refers to mutations/bp/cell division. Mutation frequency means mutations/bp. As here mutations per bp were determined, it is recommended that throughout the manuscript the term "mutation rate" is exchanged by "mutation frequency", because that is what was determined here.

b) Lines 115 and 116: The abbreviations CLL and MCL have already been introduced earlier (line 70). Introduce only when first mentioned in the main manuscript and then use abbreviations consistently.

c) Line 147: Replace "range" by "ranging".

d) Line 163:  The sequencing reaction is not a PCR, as only one primer is used (linear DNA synthesis, not exponential), Therefore, after the sequencing cycling no PCR product is obtained.

e)  Line 174: Please explain that the IGH Somatic Hypermutation assay is the approach where VH leader primers are used.

f) Lines 196 and 197: Abbreviations IGH and SHM already introduced earlier.

g) Line 225: correct "cu-off"

h) Line 238: "comparison" instead of "compare".

i) Table 2: chronic lymphocytic leukemia is now officially termed CLL. B-CLL is no longer used (because T-CLL does no longer exist as a term, so it is clear that CLL is a B cell malignancy.)

j) Line 249: Table 2 is meant here, not Table 1.

k) Line 308: I did not find where the abbreviation LT (Lymphotrack?) was introduced.

l) In the published version, the quality of the figures needs to be improved. The text in the figures is difficult to read and unsharp.

See my minor points in the report tob the authors

Author Response

(The authors gave the same response as above.)

Round 2

Reviewer 1 Report

The manuscript improved after the peer review process and in my opinion is now suitable for publication in Cancers journal

Author Response

Thank you

Reviewer 2 Report

The recommendations in my original reports were nearly completely adequately adressed.

A remaining minor point is that the term "mutation rate" was consistently replaced by "mutation frequency" in the main text, as recommended, but this exchange was forgotten in several figures and figure legends and in Table 4.

Author Response

We apologize for the inconvenience. We modified as requested